# Insights of Expression Profile of Chemokine Family in Inflammatory Bowel Diseases and Carcinogenesis

**DOI:** 10.3390/ijms251910857

**Published:** 2024-10-09

**Authors:** Yinjie Zhang, Yue Jin, Yanjing Wang, Siyi Wang, Yuchen Niu, Buyong Ma, Jingjing Li

**Affiliations:** 1Engineering Research Center of Cell & Therapeutic Antibody, School of Pharmacy, Shanghai Jiao Tong University, Shanghai 200240, China; zhyj@sxu.edu.cn (Y.Z.); jinyueyaoyao@163.com (Y.J.); wangyanjing@sjtu.edu.cn (Y.W.); wangsiyi_sjtu@163.com (S.W.); nyc123@sjtu.edu.cn (Y.N.); mabuyong@sjtu.edu.cn (B.M.); 2Modern Research Center for Traditional Chinese Medicine, The Key Laboratory of Chemical Biology and Molecular Engineering of Ministry of Education, Shanxi University, Taiyuan 030006, China

**Keywords:** inflammatory bowel disease (IBD), colon cancer, chemokines, expression profile

## Abstract

Chemokines are integral components of the immune system and deeply involved in the pathogenesis and progression of inflammatory bowel disease (IBD) and colorectal cancer (CRC). Although a considerable amount of transcriptome data has been accumulated on these diseases, most of them are limited to a specific stage of the disease. The purpose of this study is to visually demonstrate the dynamic changes in chemokines across various stages of bowel diseases by integrating relevant datasets. Integrating the existing datasets for IBD and CRC, we compare the expression changes of chemokines across different pathological stages. This study collected 11 clinical databases from various medical centers around the world. Patients: Data of patient tissue types were classified into IBD, colorectal adenoma, primary carcinoma, metastasis, and healthy control according to the publisher’s annotation. The expression changes in chemokines in various pathological stages are statistically analyzed. The chemokines were clustered by different expression patterns. The chemokine family was clustered into four distinct expression patterns, which correspond to varying expression changes in different stages of colitis and tumor development. Certain chemokines and receptors associated with inflammation and tumorigenesis have been identified. Furthermore, it was confirmed that the 2,4,6-trinitrobenzenesulfonic acid (TNBS)-induced colitis model and the azoxymethane (AOM)/ dextran sulfate sodium (DSS)-induced colon cancer model shows stronger correlations with the clinical data in terms of chemokine expression levels. This study paints a panoramic picture of the expression profiles of chemokine families at multiple stages from IBD to advanced colon cancer, facilitating a comprehensive understanding of the regulation patterns of chemokines and guiding the direction of drug development. This study provides researchers with a clear atlas of chemokine expression in the pathological processes of inflammatory bowel disease and colon cancer.

## 1. Introduction

Chemokines are a class of cytokines that gradually evolve during the development of the central nervous system in chordates [1]. In humans, this family, according to the conserved cysteine residues, is subdivided into four subfamilies, CXC, CC, CX3C, and XC, with a total of 39 members. Similarly, their receptors are also classified into four subfamilies based on the ligands they bind to, encompassing 22 genes. The chemokine system, constituting the core regulatory mechanism for directed cell migration, plays an indispensable role in normal embryonic development and immune system function.

The chemokine system, as a key mechanism, plays a crucial role in mediating the infiltration of inflammatory cells [2]. Inflammatory bowel diseases (IBD), especially ulcerative colitis (UC) and Crohn’s disease (CD), are the most common intestinal diseases closely related to autoimmune defects. Significant changes in chemokine expression levels have become one of their important pathological characteristics [3]. It is noteworthy that chemokines play an active role in bacterial colitis, but they often become a factor aggravating the disease in IBD. Therefore, chemokines are regarded as important potential targets for colitis drug development [4,5].

Colon cancer is the third most common cancer type and the second leading cause of cancer death globally. There is a causal relationship between the occurrence of some colon cancers and a history of colitis. Chemokines serve as crucial regulators in the microenvironment of colon cancer, with the presence of two opposing forces, tumor-promoting and tumor-destructive chemokines [6,7], and the balance between them represents an important factor influencing prognosis. Targeting chemokines is one of the key directions in immunotherapy [8].

Based on the information provided above, the progression from colitis to primary tumor, and further to metastatic tumor, represents a continuous process that involves several stages, including inflammation, tumor immunity, immune escape, and tumor metastasis. The correlation between the expression and silencing of chemokines and pathological stages during this process holds significant importance for understanding and treating the disease [9]. Currently, there is a lack of comprehensive studies analyzing the expression profiles of chemokines throughout this process.

In this article, we systematically integrated multiple transcriptome data resources currently available on colitis, colonic adenoma, colorectal cancer, and distant metastatic cancer, and deeply explored the expression profile information of the chemokine family genes within them. By analyzing these data, we successfully identified a set of genes that are specific to inflammation, tumorigenesis, and metastasis. The mapping of this chemokine expression atlas not only presents a clear picture of the role of chemokines in the development and progression of colorectal cancer, but also provides powerful directional guidance for further elucidating the relevant mechanisms, exploring novel therapeutic targets, and driving drug development.

## 2. Results

### 2.1. Chemokine Family Exhibits Distinct Expression Patterns in Colonic Diseases

We integrated the expression change data of chemokine genes at specific disease stages from each dataset (Appendix A) and presented them in the form of a heatmap (Figure 1A). Each color block in the figure represents the ratio of significant change in a gene during a specific disease course compared to healthy controls, with red indicating upregulation and green indicating downregulation. Based on the change trends, we clustered the chemokine ligands into four categories, which will be elaborated in detail later.

In addition, we exhibited the expression levels of chemokines in IBD and tumor tissues through a scatter plot (Figure 1B). The average change in gene expression across multiple IBD datasets was taken as the x-axis, while the average change in chemokine expression across multiple adenoma and adenocarcinoma datasets was used as the y-axis. We established a two-fold change level as the criterion for grouping, dividing the chemokine family into four groups (group i, ii, iii, iv). These four groups correspond to the four clusters in Figure 1A.

Based on the average changes in gene expression from a representative dataset GSE4183, a line chart was plotted to visually illustrate the overall trend of gene expression levels in the four groups (Figure 1C).

Next, we will describe the characteristics of the expression profiles of these four groups individually.

(i) Neoplasm-upregulated chemokines: This group of chemokines includes *CXCL1-*
*8*, *11*, and *CCL20*, *24*. These chemokines are characterized by a significant increase in expression levels during colitis, and their expression remains high even after transitioning into the tumor stage (Figure 1A). Among them, ELR+ chemokines belong to this group, with CXCL8, CXCL3, and CXCL1 exhibiting the largest changes in expression levels.

(ii) Inflammation-specific chemokines: This group includes *CXCL9*, *10*, *16*, *CCL2*, *3*, *4*, *11*, *18*, *22.* Compared to chemokines positively correlated with cell proliferation, these factors are significantly upregulated only during immune–inflammatory diseases. However, as the disease progresses to adenoma or later stages, their expression levels revert to levels close to those in healthy controls, displaying inflammation-specific expression patterns. Among them, *CXCL9* and *CXCL10* exhibit the most significant changes (Log_2_FC > 3).

(iii) Chemokines unrelated to colonic diseases: These chemokines belong to the remaining family members that do not show significant changes in expression levels, either in inflammatory bowel diseases or in tumor-related tissues.

(iv) Neoplasm-downregulated chemokines: This group of chemokines includes *CXCL12*, *13*, *14*, *CCL5*, *8*, *13*, *14*, *15*, *19*, *21*, *23*, *XCL1*, *2*. In contrast to factors positively correlated with proliferation, the expression levels of these chemokines are significantly downregulated in colorectal tumors. However, they do not exhibit significant changes during the IBD stage, demonstrating a pure tumor proliferation-specific negative correlation. Among them, *CXCL12* exhibits the most significant change (Log_2_FC < −3).

### 2.2. Changes in the Expression of Chemokines in Metastatic Cancer

We compared the expression differences between metastatic cancer and primary cancer and found that, apart from a significant downregulation of *CXCL14* and *CCL28*, as well as a significant upregulation of *CXCL12* and *CCL2*, the expression levels of most chemokines did not achieve a twofold change (Figure 1A,B, Appendix A). *CXCL14* gradually decreased throughout the disease process, with a particularly sharp decline in its expression level in metastatic tumor tissues (Figure 2A), making it rather exceptional among the entire chemokine family. *CCL28* gene data were only available in the Sidra-LUMC dataset, revealing a significant downregulation in expression. *CXCL12* gene expression was drastically downregulated in primary colon cancer (Log_2_FC = −3.1), but its expression level increased in metastatic cancer vs. primary, Log_2_FC = 1.03. *CCL2* expression, on the other hand, was mildly downregulated in primary colon cancer but increased in metastatic cancer (Log_2_FC = 1.06 ± 0.36). These findings suggest that these genes play a role in the process of tumor metastasis.

### 2.3. Changes in Receptor Gene Expression

We compared the changes in the expression levels of chemokine receptors during the colitis-to-colon cancer process (Figure 3A, Appendix A) and found that *CXCR2* (Log_2_FC = 2.7 ± 1.2) and CCR1 (Log_2_FC = 1.36 ± 0.28) were significantly upregulated in all colitis databases, consistent with the expression trends of their ligands, *CXCL1, 2, 3, 5, 6, 7, 8,* and *CCL3, 4*. In contrast, *CCR2* and *CXCR5* were significantly downregulated during the colonic adenoma stage, with *CCR2* (Log_2_FC = −1.02 ± 0.79) and *CXCR5* (Log_2_FC = −1.42 ± 0.57). No significant changes were observed in the expression levels of other receptor genes. Figure 3B depicts the expression changes of the significant genes in the colitis, adenoma, and adenocarcinoma stages from dataset GSE4183, and the trends are consistent with the results from the integration of multiple datasets.

### 2.4. Correlation between Mouse Models of Colonic Diseases and Clinical Data

The commonly used mouse models for inflammatory bowel disease are acute colitis models induced by dextran sulfate sodium (DSS) or 2,4,6-trinitrobenzenesulfonic acid (TNBS). Based on the homology between humans and mice (see Appendix A), we analyzed the correlation between changes in chemokine levels in these two mouse colitis models (Table 1) and those in human immune-related bowel disease samples (Figure 4).

The results revealed that there was a strong correlation between changes in chemokines in both the TNBS and DSS models and those in human IBD (Figure 4A,B, Appendix A). In the TNBS model, the correlation coefficients for changes in chemokines, excluding *CXCL10* and *11*, reached r = 0.73 (*p* < 0.0001).

Currently, the most commonly used model is the AOM-DSS model [11], while there is also the ApcMin/+/J mouse spontaneous colon cancer model [12]. We compared the correlation between the fold change of chemokine expression in these two models (Table 1) and that in the human colon cancer clinical databases. The results showed that the trend of chemokine changes in the AOM-DSS model highly matched the clinical data (r = 0.71 and *p* < 0.0001), except for *CXCL12* and *CXCL14* exhibiting significant differences. However, the correlation between the APC (Min) spontaneous colon cancer model and the clinical data was relatively weak (r = 0.37, *p* = 0.041).

**Table 1 ijms-25-10857-t001:** The datasets used in this study.

Clinical Data						
Dataset Accession No. or Dataset No.	Sample Number					Ref.
	Control	IBD	Adenoma	Primary Carcinoma	Metastasis Carcinoma	
GSE49355	18			20	19	Del Rio et al. [13]
GSE41258	54			186	67 ^a^	Sheffer et al. [14]
GSE4183	7	15	15	15		Galamb et al. [15]
GSE37364	38		29	14	13	Galamb et al. [16]
GSE23878	24			35		Uddini et al. [17]
GDS2947 ^b^	32		32			Sabates et al. [18]
GDS4382 ^b^	17			17		Khamas et al. [19]
GDS3119	5	8				Olsen et al. [20]
GSE36807	7	15 ^c^				Montero-Melendez et al. [21]
Sidra-LUMC				69	36	Roelands et al. [10]
GSE14580	6	24				Arijs et al. [22]
Mouse Model						
Dataset	Model			Healthy	Disease	
GSE22307	DSS-induced UC model			5	5	Fang et al. [23]
GSE13705	TNBS-induced IBD model			3	3	Billerey-Larmonier et al. [24]
GSE31106	AOM-DSS-induced CAC model			3	3	Tang et al. [25]
GSE43338	C57BL/6J-ApcMin/+/J spontaneous CRC			3	5	Neufert et al. [26]

^a^ 47 liver metastasis and 20 lung metastasis; ^b^. paired samples; ^c^. only UC were included.

## 3. Discussion

IBD is a significant intestinal disease, and chemokine-mediated immune cell infiltration is one of its major pathological processes [27]. CRC is the third most diagnosed cancer and the second leading cause of cancer death worldwide, and chemokine-mediated regulation of the tumor microenvironment plays a crucial role in patients’ prognosis. There is a certain causal relationship between IBD and CRC [7], and studying them as a whole holds significant theoretical and practical implications for understanding the pathogenesis of colorectal cancer.

The highlight of this study is the collection and comprehensive analysis of existing multiple transcriptomics data on the chemokine family across different pathological stages of colitis–colorectal cancer, resulting in a continuous expression profile. The entire process is divided into four stages: immunological bowel disease, adenoma, primary colorectal cancer, and metastatic cancer, with each stage encompassing three to six datasets. This approach primarily serves to avoid statistical biases in the expression patterns of certain genes that may arise from the random error of a single dataset. More importantly, there is currently no single dataset that covers all there pathological stages. By integrating various scattered datasets, we have obtained a comprehensive dynamic picture of chemokine expression levels.

This study yielded several significant results. Firstly, chemokines during the IBD stage can be clearly divided into two groups, with one group showing significant upregulation, while the other group remains unchanged comparing to the controls. This finding holds significant importance for understanding the occurrence mechanisms of colitis and for drug development. Analysis of the expression profile reveals that numerous chemokines are involved in IBD, potentially indicating a degree of redundancy among them. Therefore, antagonizing a single chemokine may not produce significant therapeutic effects. For example, Eli Lilly developed a neutralizing antibody, Eltrekibart (LY-3041658), targeted against ELR+ chemokines for the treatment of inflammatory bowel disease [28]. This antibody is capable of simultaneously antagonizing the activities of seven chemokines with the ELR motif, namely *CXCL1*, *2*, *3*, *5*, *6*, *7*, and *8* (their corresponding receptors are *CXCR1* and *CXCR2*). However, the study was terminated due to undisclosed reasons, and the authors speculate that it may be because the antibody did not achieve the expected therapeutic effect. Given the upregulation of numerous chemokines in IBD, it may be necessary to consider simultaneously interfering with multiple pathways to overcome the functional redundancy of chemokines.

Secondly, the genes that undergo significant changes in colon cancer can be classified into upregulated and downregulated groups. We defined the genes in the former group as neoplasm-upregulated chemokines (NUCs, Figure 1), where all ELR + chemokines are clustered. The latter group was referred to as neoplasm-downregulated chemokines (NDCs). According to the clonal selection theory of cancer cells, the expression regulation of NUCs and NDCs in tumors may represent clonal traits that are fixed and facilitate tumor proliferation or immune escape. Intervening in these traits may become a potential approach for the treatment of colon cancer.

*CXCL14* was clustered within the neoplasm-downregulated chemokines cluster, but its sharp downregulation during tumor metastasis particularly stands out (Figure 3). It is the only gene that is significantly downregulated in all datasets comparing metastatic cancers to primary cancers, with an average downregulation fold change of Log2FC = 2.3 ± 0.7. This suggests that *CXCL14* silencing plays an important role in the process of colon cancer metastasis, as mentioned in a previous study [29].

Compared to the drastic changes in ligand expression, the expression of receptor genes does not exhibit remarkable variations. Only four receptors, namely *CXCR2*, *CCR1*, *CXCR5*, and *CCR2*, showed significant changes in over half of the datasets in a certain pathological stage. Among them, *CXCR2* was drastically upregulated in the IBD and adenoma stage, which is corroborated by the general upregulation of its ligands, *CXCL1-3* and *CXCL5-8* (Figure 1). This suggests that immune cells related to the CXCR2 axis are deeply involved in these pathological processes. Previous studies have shown that neutralizing *CXCR2* can inhibit neutrophil-mediated colitis [30]; furthermore, genetic knockout of the *CXCR2* gene is capable of inhibiting the formation of primary colon cancer in mice [31]. Similarly, *CCR1*, as a common receptor for numerous CCL ligand family members (*CCL3*, *CCL5*, *7*, *8*, *CCL13-16*, *CCL23*), also exhibits an extremely significant upregulation in the IBD stage. However, interestingly, among these ligands, only *CCL3* shows a significant upregulation in IBD (Figure 1), indicating that the CCL3-CCR1 axis plays a crucial role in this pathological process, which is consistent with previous studies [32]. In addition, there are two other significantly downregulated receptor genes, *CXCR5* and *CCR2*, which both show a marked downregulation in adenoma samples. This downregulation echoes the corresponding decrease in their respective ligands, *CXCL13* and *CCL2*. Previous studies have indicated that the CXCL13-CXCR5 axis is related to the prognosis of colon cancer [33]. *CCL2* is downregulated in adenoma but significantly upregulated in metastasis samples (Figure 2), which aligns with previous findings that it promotes liver metastasis [34,35].

Finally, appropriate animal models are of great significance for drug development in preclinical studies. Given that chemokine-targeted drug development necessitates the utilization of suitable mouse models, we examined the correlation between chemokines in various mouse models of IBD and primary colon cancer and clinical samples of human colon cancer. Through the comparison of correlation coefficients, we determined that colitis induced by TNBS chemical challenge and colon cancer generated through the AOM-DSS protocol exhibit closer chemokine expression profiles to clinical samples (Figure 4). As a result, we may prioritize these two models in future research endeavors.

This study also has its limitations. Firstly, we identified genes with significantly different expression levels by using the criteria of both a *t*-test Q-value < 0.05 and an expression change greater than two-fold (|Log_2_FC | > 1). However, there are some genes that own a | Log_2_FC | < 1 and have a Q-value significantly less than 0.05. The correlation of these genes with the disease may have been underestimated in this study. Readers interested in this aspect can refer to the Appendix A for detail data.

Secondly, to make the results simpler and more readable, we treated Crohn’s disease and ulcerative colitis collectively as IBD in our study. Although Crohn’s disease and ulcerative colitis share some similarities, they are two distinct diseases with different causes. Additionally, we combined liver metastasis and lung metastasis samples as metastatic samples for analysis. These processing procedures may have overlooked some disease-specific genes.

In conclusion, this study has drawn a panoramic picture of the expression profiles of chemokine families at multiple stages from colitis to advanced colon cancer, through the integration of transcriptomics data. These findings provide researchers with a more macro-level perspective, facilitating a comprehensive understanding of the regulation patterns of chemokines and guiding the direction of drug development.

## 4. Materials and Methods

### 4.1. Data Collection

This study comprehensively integrated multiple datasets encompassing colitis, colonic adenomas, colonic primary tumors, and metastatic tumors (as shown in Table 1), aiming to delve into the critical molecular mechanisms underlying the occurrence and progression of the disease. During the data processing, we focused on extracting the expression profile data of chemokine genes and carefully summarizing them, as detailed in Appendix A. It needs to be emphasized that in cases where multiple probes existed for certain genes, we adopted the probe with the strongest signal as valid data for subsequent analysis (Appendix A).

### 4.2. Data Analysis

Across different datasets, publishers provided varying annotations for the same type of samples. Based on the annotations provided by the dataset publishers regarding the disease stage of the samples, the data are categorized into five primary groups: (1) healthy controls, including samples annotated as “none”, “control”, “normal”, and “normal colonic mucosa”; (2) IBD, encompassing samples annotated as “IBD”, “UC”, and “CD”; (3) colon adenoma; (4) primary colon cancer, comprising samples annotated as “colorectal cancer”, “primary tumor”, and “carcinoma”; (5) metastatic cancer, including samples annotated as “metastatic cancer”, “liver metastasis”, and “lung metastasis”.

To ensure the comparability of the data and the accuracy of the analysis, we converted the linear numerical values in the dataset to logarithmic values (Log_2_), since the log-converted data are closer to a normal distribution. For the differential analysis, we selected the appropriate t-test method based on the source and type of samples. For paired data, which refers to samples taken from different tumor stages of the same patient, we employed the paired t-test to reveal the dynamic changes in gene expression during disease progression. For unpaired data, we used the unpaired Student t-test to assess the differences in gene expression between different disease groups. In all t-tests, we assumed equal variance and adopted a two-tailed test. Additionally, to control the false positive rate, we also applied the Benjamini–Hochberg method to transform the *p*-values to Q-values. Statistical significance: *: *p* < 0.05, **: *p* < 0.01, ***: *p* < 0.001, ****: *p* < 0.0001.

### 4.3. Graphical Representation

Volcano Plot: We created a volcano plot to visually illustrate the changes in gene expression. In the volcano plot, the horizontal axis represents the mean difference in gene expression (denoted as Log_2_FC), while the vertical axis employs the -Log_10_Q value. This setup facilitates the rapid identification of genes with significant differences, specifically those that demonstrate both significant changes in expression levels and high statistical reliability.

Heatmap: To further analyze gene expression patterns in depth, we generated a heatmap. In creating the heatmap, we established strict screening criteria: only genes with a -Log_10_Q value greater than 1.3 (i.e., a Q-value < 0.05) and an | Log_2_FC | > 1 were considered to have a significantly different expression. Based on these criteria, genes with a significantly increased expression were assigned a score of 1, while genes with a significantly decreased expression were assigned a score of −1. Genes without significant changes were scored as 0. Subsequently, we calculated the arithmetic mean of scores across different datasets for the same disease stage, yielding a final expression change score for each gene at that stage. For instance, CCL20 exhibits a significant increase (LogFc > 2, Q-value < 0.05) in 3 out of 4 IBD databases; we define the “ratio of significant difference” as 3/4 = 0.75 in IBD stage. This value was then used to construct the heatmap. Finally, we used SPSS software (v25) to perform a clustering analysis on the gene score matrix, generating a heatmap that visually displays the gene expression patterns (as shown in Figure 1A).

Scatter Plot: In the process of creating the scatter plot, we first calculated the mean fold change (FC) of every gene across the IBD datasets, which served as the horizontal axis. Subsequently, we computed the average gene variation for both the adenoma dataset and the adenocarcinoma dataset, and then took the arithmetic average of these two means as a representative of the gene changes in the tumor, which was plotted on the vertical axis. Through this approach, we were able to compare and visualize the differences and trends in gene expression changes between IBD and tumor stages (as depicted in Figure 1B).

Line chart of gene expression: To systematically demonstrate the changes in gene expression across different pathological stages within each dataset, we have also constructed a line chart that tracks gene variation. In these line charts, the arithmetic means of gene variation for each dataset at different pathological stages were calculated. By connecting the points, we clearly illustrated the dynamic process of gene expression changes over the course of the disease progression.

## Figures and Tables

**Figure 1 ijms-25-10857-f001:**
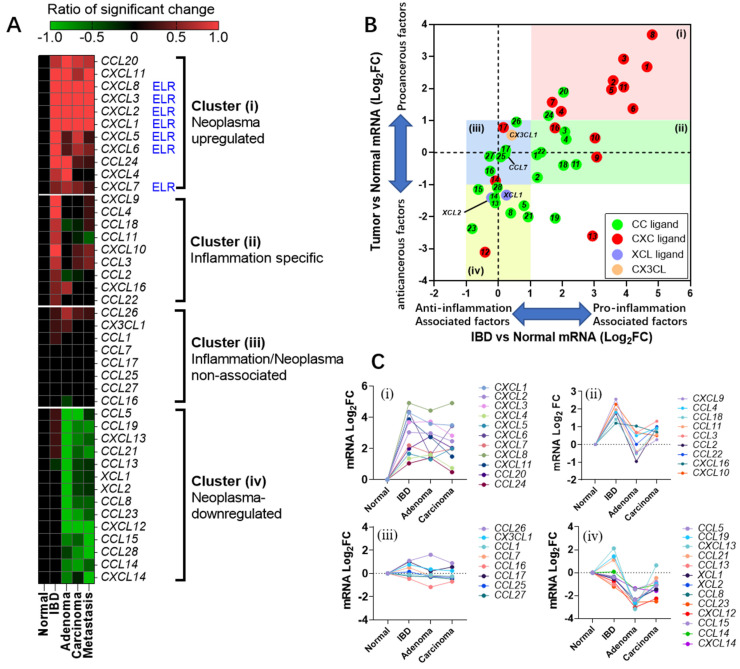
Expression profiles of the chemokine family in the pathological process of colitis to colon cancer. (**A**) Heatmap of expression changes. The percentage of significant changes was obtained by dividing the number of datasets with significant changes in expression levels at different pathological stages compared to healthy colon tissue by the total number of datasets in the database. The chemokine family was clustered into four groups based on the trends of expression changes. ELR: ELR motif + chemokine. (**B**) A scatter plot was created with the expression changes of chemokines in the IBD datasets as the x-axis and the average changes in the adenoma and adenocarcinoma datasets as the y-axis. Chemokines were divided into four groups (i, ii, iii and iv) based on the cut-off values of +1 and −1 for Log_2_FC in the two dimensions. The genes in these four groups correspond to the four clusters in (**A**). (**C**) Line graphs of expression changes of genes in the four clusters in (**A**) were illustrated during the colitis to colon cancer process.

**Figure 2 ijms-25-10857-f002:**
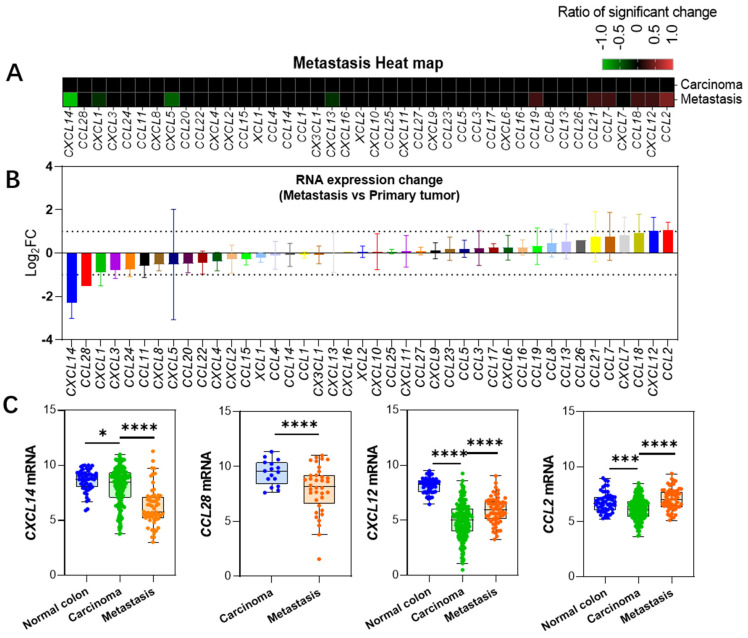
Changes in gene expression during metastatic cancer progression. (**A**) Heatmap showing alterations in gene expression levels in metastatic cancers developed from primary colon cancer. (**B**) Changes in expression levels relative to primary colon cancer in metastatic cancers, presented as mean ± SD based on multiple datasets. Note that *CCL26*, *CCL28*, and *CXCL16* are only present in one dataset, hence no standard deviation data is available. (**C**) Expression level data for four genes showing significant changes. Data for *CXCL14*, *CXCL12*, and *CCL2* are derived from the GSE41258 dataset, while *CCL28* data is from the Sidra-LUMC dataset [10]. *: *p* < 0.05, ***: *p* < 0.001, ****: *p* < 0.0001.

**Figure 3 ijms-25-10857-f003:**
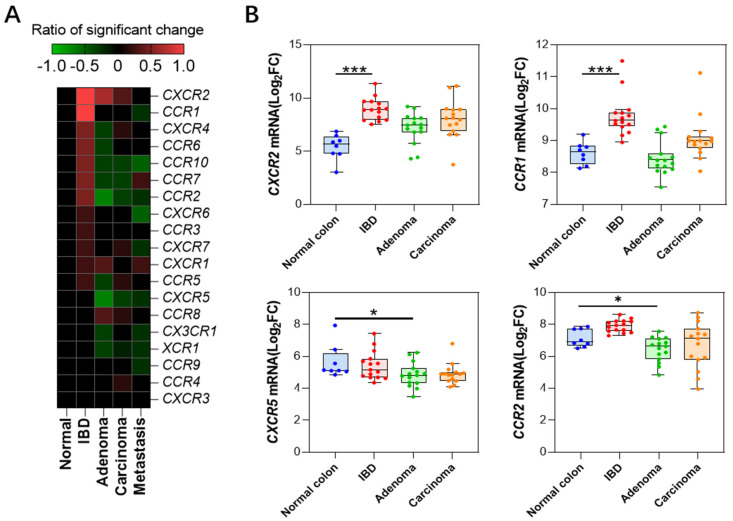
Chemokine receptor expression profiles (**A**) Heatmap depicting changes in expression levels of chemokine receptors at different stages. (**B**) Expression profiles of four receptor genes with significant changes in expression levels (GSE4183 dataset). *: *p* < 0.05, ***: *p* < 0.001.

**Figure 4 ijms-25-10857-f004:**
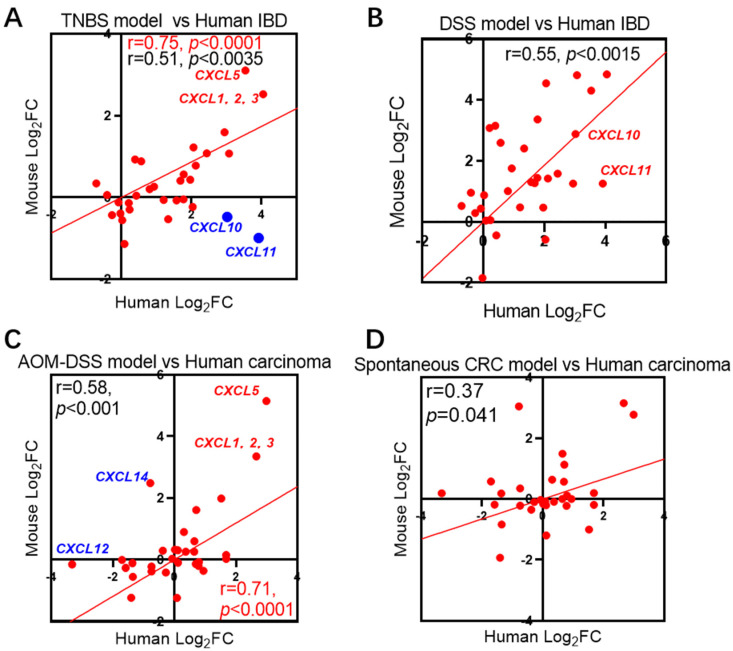
Correlation between chemokine expression changes in mouse intestinal disease models and clinical databases. (**A**) Correlation between mouse TNBS colitis model (GSE13705) and human IBD data. (**B**) Correlation between mouse DSS colitis model (GSE22307) and human IBD data. (**C**) Correlation between mouse AOM-DSS colon cancer model (GSE31106) and human colon cancer data. (**D**) Correlation between spontaneous colon cancer in ApcMin/+/J mice (GSE43338) and human colon cancer databases.

## Data Availability

Data is contained within the article and Appendix A.

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
