# Peer review of "Insights of Expression Profile of Chemokine Family in Inflammatory Bowel Diseases and Carcinogenesis"

_ijms, 2024, doi:10.3390/ijms251910857_

Round 1

Reviewer 1 Report

Comments and Suggestions for Authors

The original article titled “Insights of expression profile of chemokine family in inflammatory bowel diseases and carcinogenesis”, by the authors Yinjie Zhang, Yue Jin, Yanjing Wang Siyi Wang,Yuchen Niu Buyong Ma and Jingjing Li, is well-written with a clear objective, emphasized in the title itself.

It is the result of efforts to get a comprehensive picture of the changes in the chemokine expression patters in various stages of carcinogenesis of colorectal cancer – from precancerous states (IBD) over tumor development to the metastasis. The fact that this study considers a continuum of molecular processes and their changes, is its main quality and offers a fresh perspective in the diagnosis and in the drug design of CRC.

However, there are certain issues to be addressed:

Major issues:

- In the text the term “multi-omics” is mentioned several times, which is not completely accurate since all the data refer to the expression profiles of various genes coding either chemokine molecules or their receptors. Thus, the term transcriptomics would be more appropriate.

- in the material & methods section, although the generation and types of graphs are explained in detail, nothing is said about which medical centers were contacted in order to obtained data and how ethical issues were managed in order to protect the data used

- in the discussion section, not all figures are cited even when discussed (see minor issues), and the discussion of chemokine receptor expression profiles (figure 3) is missing completely

- given the fact that the topic of the study is very attractive, with a vast literature published recently, the number of only 29 reference entries is far too small

Minor Issues:

Line 280 - “in certain genes” should be “in the expression patterns of certain genes”

Line 283 - “of chemokines” should be “of chemokine gene expression levels”

Line 286 - “remains unchanged” should be “remains unchanged comparing to the controls”

Line 308 - figure 2 should be mentioned

Line 318-320 - figure 4 should be mentioned

Comments on the Quality of English Language

minor editing of English language is required

Author Response

Major issues:

- In the text the term “multi-omics” is mentioned several times, which is not completely accurate since all the data refer to the expression profiles of various genes coding either chemokine molecules or their receptors. Thus, the term transcriptomics would be more appropriate.

Response:Thanks for the reviewer’s comments, the term “multi-omics” were all corrected into “transcriptomics”.

- in the material & methods section, although the generation and types of graphs are explained in detail, nothing is said about which medical centers were contacted in order to obtained data and how ethical issues were managed in order to protect the data used

Response: The original of the data are all from the public database. The detailed information, such as medical centers, could be found in references in table 1.We conduct our research in strict adherence to the guidelines set forth by the Clinical Data Ethics Committee of Shanghai Jiao Tong University.

- in the discussion section, not all figures are cited even when discussed (see minor issues), and the discussion of chemokine receptor expression profiles (figure 3) is missing completely

 Response: We have discussed the expression profile of chemokine receptors in discussion section (after line 314), and add several references.

- given the fact that the topic of the study is very attractive, with a vast literature published recently, the number of only 29 reference entries is far too small

 Response:After revision, the references number reached to 35.

Minor Issues:

Line 280 - “in certain genes” should be “in the expression patterns of certain genes”

Line 283 - “of chemokines” should be “of chemokine gene expression levels”

Line 286 - “remains unchanged” should be “remains unchanged comparing to the controls”

Line 308 - figure 2 should be mentioned

Line 318-320 - figure 4 should be mentioned

Response:

Thanks for reviewer’s rigorous comments, the all minor issues were revised accordingly.

Reviewer 2 Report

Comments and Suggestions for Authors

This manuscript presented systematically integrated multiple transcriptome data resources currently available on colitis, colonic adenoma, colorectal cancer, and distant metastatic cancer, and deeply explored the expression profile information of the chemokine family genes within them.

I have several concerns related to this manuscript.

1. This study is more like a review, not the original article.

2. Methods used in this study are unclear

3. The manuscript requires editing changes as the abstract, citation, and structure of the text do not fully match the requirements of the IJMS journal.

4. The language of this manuscript should be revised, e.g., wording like "It is worth noting that" should not be placed in the result section as this is not the scientific wording in this part of the text.

5. There is limited rationale for this study. What does this study add? What is the difference between this study and other studies? Why this study is important? 

6. The discussion is too short and does not fully match the scope of the study.

Author Response

    1. This study is more like a review, not the original article.

    Response: I would like to express my sincere gratitude to the reviewers for their valuable comments. It is important to clarify that this article does not constitute a review but rather presents an original study within the field of bioinformatics. Through meticulous effort, we have extracted expression profile data of chemokines from a comprehensive collection of databases. We have subsequently integrated these data across various case stages, enabling us to meticulously map the dynamic profiles of genes belonging to the chemokine family. This approach provides unique insights into the complex dynamics of chemokine expression.

    1. Methods used in this study are unclear

    Response: Thank you for the reviewer's thoughtful comments. In response, we have thoroughly revised the Methods section to ensure clarity and accuracy. Should there be any remaining aspects that remain unclear, we kindly request that you specify the relevant paragraph or section, and we will promptly attend to further revisions as necessary.

    1. The manuscript requires editing changes as the abstract, citation, and structure of the text do not fully match the requirements of the IJMS journal.

    Response: Thank you for the reviewer's invaluable comments. We are committed to ensuring that our manuscript meets the highest standards set by the journal, and as such, we will meticulously revise it to comply with all the specific formatting requirements according to editor’s guidance.

    1. The language of this manuscript should be revised, e.g., wording like "It is worth noting that" should not be placed in the result section as this is not the scientific wording in this part of the text.

    Response: The phrase "It is worth noting that" has been revised to "It needs to be emphasized" for greater emphasis. Additionally, the entire manuscript has undergone a thorough review by a native English speaker to ensure clarity, fluency, and adherence to academic writing standards.

    1. There is limited rationale for this study. What does this study add? What is the difference between this study and other studies? Why this study is important? 

        Response: Thank you for the reviewer's insightful comments. As emphasized in our revised abstract, this study uniquely integrates the dynamic expressions of chemokine families across the spectrum of colitis, extending from primary colon cancer to metastatic cancer. Our primary focus lies in elucidating the hitherto unexplored dynamic changes of chemokines, which offer a novel perspective. Our aim is to present a comprehensive view of colitis and colon cancer, thereby fostering fresh avenues for drug development and therapeutic strategies.

    1. The discussion is too short and does not fully match the scope of the study.

    Response: We have added a paragraph about the receptor expression profile discussion (Line315~335).

Round 2

Reviewer 1 Report

Comments and Suggestions for Authors

Thank you for the responses and changes.